# Exploring the Impacts of Exit Structures on Evacuation Efficiency

**Xiaoge Wei** [1,2,*] , **Zhen Lou** [1,2,*], **Huaitao Song** [1,2] , **Hengjie Qin** [1,2] and **Haowei Yao** [1,2]

1   The School of Building Environment Engineering, Zhengzhou University of Light Industry, Zhengzhou 450001, China; songhuaitao@zzuli.edu.cn (H.S.); qhj102432@zzuli.edu.cn (H.Q.); yaohaowei@zzuli.edu.cn (H.Y.)
2   Zhengzhou Key Laboratory of Electric Power Fire Safety, Zhengzhou University of Light Industry, Zhengzhou 450001, China
*   Correspondence: 2015041@zzuli.edu.cn (X.W.); loganlz@zzuli.edu.cn (Z.L.)

**Abstract:** In the context of a fire emergency, safe and efficient exits are of paramount importance for pedestrian evacuation. The recent rapid development in the construction industry has rendered exit structures more diverse and complex. However, little attention has been paid to the influence of exit structures on the efficiency of crowd evacuation processes. In this paper, a tentative experiment was designed to preliminarily reveal the effects of five exit structures (Exit 1, Exit 2, Exit 4, Exit 5, and Exit 3 as examples for comparison) on crowd evacuation. Exit 1 has door leaves opening outward. Exit 2 has door leaves opening inward. Exit 3 has no leaves. Exit 4 consists of double-layer exit doors with the doors opening outward. Exit 5 comprises double-layer exit doors with the doors opening both sides outwards Subsequently, according to the properties of this experiment, a social force-based simulation model was established using the AnyLogic software 8.8.4. By changing the exit width and the crowd density, data such as evacuation time, flow rate, crowd density, and time delay were investigated in detail. The results revealed a notable variation in the evacuation efficiency depending on the deign of the exit. The respective flow rates for Exits 1, 2, 3, 4, and 5 were 0.66 people/(m·s), 0.77 people/(m·s), 0.80 people/(m·s), 0.71 people/(m·s), and 0.66 people/(m·s). Although Exit 3 excelled in terms of evacuation efficiency, it is not directly applicable to real architectural structures. Therefore, Exit 2 emerged as a highly promising solution in terms of flow rate and population control in the exit area, underscoring the effectiveness and practicality of its structural design. It is prospective that the results of this study can offer engineering and technical professionals valuable references and guidance concerning the design of exit structures.

**Keywords:** exit structure; evacuation efficiency; evacuation time; flow rate; AnyLogic software

## 1. Introduction

During a fire incident, the establishment of a secure and efficacious evacuation system is imperative to ensure the seamless evacuation of individuals. Exits serve as the final step in buildings' evacuation systems, and their traffic capacity plays a crucial role in the overall evacuation efficiency within the building. The abrupt reduction in spatial structures, such as building entrances and exits, door panels, turnstiles, and diversion devices, often constrains pedestrian flow and can easily lead to congestion during evacuation. This, in turn, results in localized high-density crowds and constitutes a significant factor of stampede accidents.

Amid a fire evacuation, a large number of people suddenly rush to the exits, causing congestion and queuing in the exit area, making it a bottleneck in the evacuation system, and increasing the difficulty of evacuation [1]. The factors influencing the evacuation capacity of building exits can generally be divided into two aspects: the characteristics of a crowd's movement behavior and the interactions of the crowd, and the environmental conditions of the building's exits. Many scholars pay more attention to pedestrians' behavior when choosing which exits to use [2]. For example, Kinateder et al. [3] investigated the effects of exit familiarity and the egress behavior of other pedestrians on exit choice in a virtual ambulatory environment. They found that the evacuation behavior of individuals

was influenced by their familiarity with the exits and the behavior of the people around them, and that this social influence increased with the size of the crowd. Furthermore, they found that these tendencies depended on an interaction between social influence and the affordances (means of egress) of the built environment [4]. Haghani et al. [5] carried out evacuation experiments to study pedestrians' exit choices and exit choice adaptation under different levels of simulated urgency. Lovreglio et al. [6] adopted VR experiments and discrete choice models to investigate how people select which exits to use in built-environment disasters. Their results show that exit decisions made during evacuation are affected by multiple factors such as social influence, the distance between an individual and an exit, the presence of smoke, and exit familiarity. Edrisi et al. [7] compared three different exit choice models: a shortest-path exit choice, a multinomial logit model, and a modified multinomial logit model with revising decisions. Their results indicate that the modified exit choice model outperforms the two other models due to its realistic representation of human behavior. Tsurushima et al. [8] focused on the disruption of symmetry when choosing exits from the viewpoint of herding, a cognitive bias seen in humans during disaster evacuations. Their simulation with the evacuation decision model showed that almost all individuals gather at one exit with some frequency, despite the individuals choosing their exit randomly. Gaire et al. [9] proved that there are significant differences in exit choices between individuals with disabilities and individuals without disabilities.

Meanwhile, lots of scholars have researched the effect of an exit's structure on the evacuation process. Various exit condition parameters, such as the number of exits [10], width of exits [11], arrangement of exits [12], thickness of exits [13], the angles of turns in exit routes [14], exit signage [15,16], and exit location [17], directly impact the process and effects of crowd evacuation. In addition, the panic induced by a fire can potentially have a detrimental impact on the efficiency of evacuating through exits, exhibited by the "fast is slow" phenomenon. Within a certain range, the crowd evacuation duration decreases with an increase in exit quantity and width, but increases with increases in exit thickness and turning angles. When the total exit width is small, the capacity of the exit becomes the main factor affecting the total evacuation time. If the total exit width is large, the pedestrian's travel distance to the exit becomes the main factor influencing the evacuation time [18]. As the distance between two adjacent evacuation doors increases (within the range of less than two body widths), the evacuation efficiency of the crowd decreases [19,20]. Further research by scholars has shown that both excessive and insufficient distances are not conducive to evacuation [21]. Under unchanged exit conditions, evenly utilizing all exits is a general approach to enhancing the overall evacuation efficiency of the building [22,23]. Song et al. [24] proposed an exit distribution strategy based on non-linear changes in an exit's capacity, which can more precisely calculate queuing times near exits. Kurdi et al. [25] put forward a balanced evacuation algorithm for the safe evacuation of pedestrians from facilities with multiple exits.

In recent years, due to the rapid and robust development of the construction industry, the structural forms of exits have become diverse and complicated. As a result, scholars have begun to focus on the effect of exit arrangements on the evacuation process. Li et al. [26] found that compared with flat exits, convex exits are more efficient in terms of evacuation time, especially suitable for situations with a higher expected crowd speed. Ding et al. [27] explored the impact of the shape of building exit turnstiles on crowd evacuation and found out that wedge-shaped turnstiles can significantly improve evacuation efficiency. Wang et al. [28] found that setting a buffer zone in front of an exit can effectively reduce crowd density at the exit and alleviate the surgent transmission of density waves in crowds, so as to reduce the risk of crowd squeezing and trampling. In addition to the impact of the exits' structures, obstacles near exits can also influence the crowd evacuation process. The placement of obstacles in the exit area is considered an effective strategy for enhancing pedestrian flow and optimizing the evacuation process [10]. Haghani et al. [29] found that placing a panel-like barricade at exit can facilitate the outflow and reduces the egress time, but its effect depends on the widths of exit, as well as the size of the barricade and its distance to exit. The shape, size, position, and the number of such obstacles, and the presence of slow-moving pedestrians, all affect the

evacuation efficiency at the exit [30]. Pedestrian flow exhibits a non-monotonic dependence on the distance of obstacles from the door. As their distance from the exit increases, the evacuation efficiency shows a trend of initially decreasing, followed by an increase, and then a decrease again. This implies that there is an optimal position for setting obstacles in the exit area, consistent with results in numerical simulations of granular flow. When the obstacle area is the same, square obstacles have the most significant impact on emergency evacuation. Compared to single-column obstacles, panels and double-column obstacles offer more stability and can enhance pedestrian evacuation efficiency [31]. When the size of the obstacle becomes large, it can be seen as a diversion wall. If the length of the diversion wall is 1.5 to 2 times the width of the exit and is positioned at a distance of 0.5 to 0.75 times the width of the exit from the exit itself, this diversion wall can effectively alleviate or even eliminate congestion during the evacuation process [32]. The physical mechanism behind improving evacuation efficiency through obstacle placement is that effective spatial separation can significantly reduce the density levels of crowded areas. However, some scholars have raised doubts about the positive effects of exit area obstacles on crowd evacuation. Garcimartín et al. [33] conducted experiments and showed that pedestrian flow does not necessarily change due to the presence of obstacles. They also found that obstacles may weaken collective lateral movement and even lead to people falling in competitive evacuation scenarios. Moreover, when pedestrians lack a complete understanding of evacuation information, the presence of obstacles may not necessarily alter pedestrian flow [34]. To address the controversies found in the aforementioned research, systemic research on the impact of obstacles in building exit areas on crowd evacuation regulations is still needed. Additionally, the current research has not yet considered the influence of complex exit configurations (such as double-layer exits, parallel multi-door exits, etc.) on exit evacuation efficiency. Studying these aspects is of great importance for future building designs and management.

Drawing inspiration from prevalent exit structures in contemporary public buildings across China, this paper employs the AnyLogic simulation platform to establish a pedestrian evacuation model in crowd evacuation scenarios. To explore the effect of the exit structure on the exit's evacuation efficiency, a series of experiments with varying crowd densities and exit structures are carried out. This research endeavors to assess the efficiency of diverse exit configurations and obtain an optimal exit design scheme.

## 2. Materials and Methods

### 2.1. The Structure of Typical Building Exits

Exits play a crucial role in a building's evacuation system. As a bottleneck, an exit's structural characteristics and associated obstacles can influence the crowd egress capacity and efficiency. As China's urbanization process accelerates, architectural spaces are expanding and becoming more intricate. This transformation has also influenced the design of building exits. While traditional exit structures are simple, as shown in Figure 1a,b, contemporary buildings increasingly feature double-layered or multiple side-by-side exits, as depicted in Figure 1c,d. This shift is particularly noticeable in large commercial buildings.

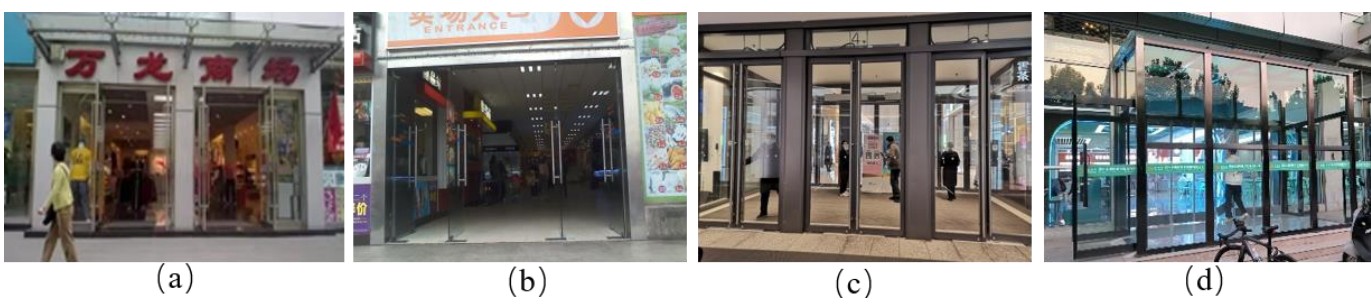

**Figure 1.** Several common exit structures of public buildings. (**a**) Doors opening outward; (**b**) doors opening inward; (**c**) double-layer exits with doors opening outward; (**d**) double-layer exits with doors opening to both sides.

These new exit structures not only improve visual aesthetics but also promote energy conservation and heat retention. However, these innovative exit designs also raise concerns about their impact on egress capacity. It is essential to consider whether these complex exit forms meet fire safety standards and requirements, and whether they facilitate or hinder evacuation procedures. These issues require in-depth investigation.

### 2.2. Introduction of the Anylogic Software

AnyLogic is a leading simulation software for various business applications that is widely employed across multiple sectors, including, but not limited to, healthcare, logistics, manufacturing, and, as discussed in this paper, crowd evacuation scenarios. The AnyLogic pedestrian library contains a range of robust modeling tools and elements designed specifically to support pedestrian dynamics within diverse systems. These tools find extensive applications in the analysis of pedestrian flows in public spaces, evacuation processes, transportation hubs, and more. Here, we report some key details about the motion model implemented in this pedestrian library. The calculation of the shortest paths for pedestrians in the pedestrian library is based on the A* algorithm [35], and the pedestrian motion is based on the social force model [36]. By using the pedestrian library, users can create sophisticated simulations to analyze and resolve congestion, optimize an area's layout, perform evacuation analyses, and more. Due to the powerful function of this software, a large number of scholars [37–39] have applied it in the study of crowd evacuation. As crowds navigate the exit area, significant interactions occur among individuals and the obstacles in their vicinity. These interactions play a pivotal role in determining the efficiency of evacuation procedures. Consequently, an evacuation model capable of precisely representing these pedestrian interactions is essential for simulating crowd evacuations in exit areas. The social force model adopted by the AnyLogic software confers a notable advantage in this regard, as it excels in capturing and modeling these complex interactions. The evacuation model of the exit area studied in this paper is shown in Figure 2.

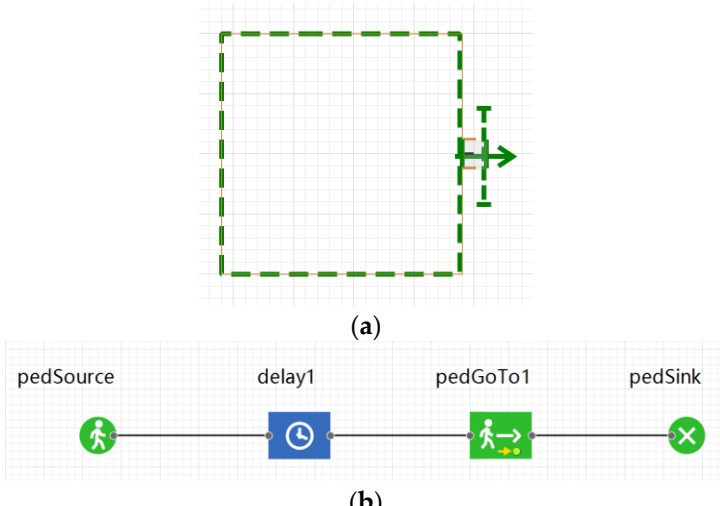

(a)

(b)

**Figure 2.** Pedestrians evacuating from the exit area. (**a**) A screenshot of the evacuation model in the exit area; (**b**) the process module of the evacuation model.

### 2.3. Simulation Setup

To assess the efficiency of various exits during an evacuation, we constructed an exit area measuring 10 m by 10 m using the AnyLogic platform. Referencing the characteristics of several building exits, as shown in Figure 1, we developed five simulation scenarios. These simulations, including a control experiment depicted in Figure 3c, were all simplified versions of the original scenarios. Figure 3a, b, d, and e correspond to the exit structures of Figure 1a, b, c, and d, respectively. The evacuation boundary is indicated with a solid green line, while the dashed line designates the point at which the evacuees have successfully

escaped. For Exits 2 and 3, these two positions coincide. Each door in our model has a thickness of 0.05 m, with variable door lengths, denoted as 'd'. The 'd' values range from 0.6 m to 1.0 m, in 0.1 m increments. For observation purposes, a density area measuring d × 2d is established at the position of the solid green line. The evacuation boundary is depicted with the solid green line, and the exit boundary is expressed by the dotted green line. The number of people within the evacuation area can be adjusted to represent different crowd densities: 100 people for a low-density crowd, 300 people for a medium-density crowd, and 500 people for a high-density crowd. In the model initialization phase, the crowd is randomly distributed throughout the exit area. An overview of these simulated experimental conditions is illustrated in Figure 2. To reduce the impact of randomness in the results, each scenario's evacuation simulation experiment was conducted three times. Ultimately, key parameters including the evacuation time, flow rate, and crowd density in the exit observation area were examined to evaluate the efficiency of different exit structures.

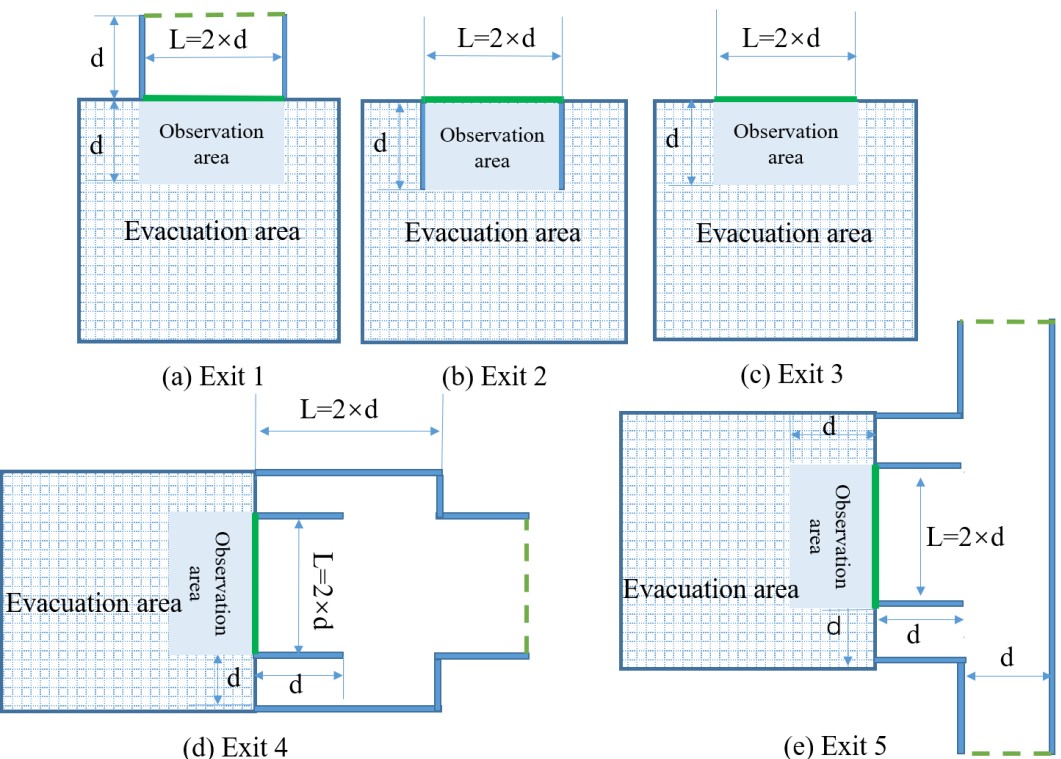

**Figure 3.** Five different exit evacuation scenarios.

## 3. Results

### 3.1. Screenshots of the Crowd Gathering at the Exit during Evacuation

Observing and recording how crowds gather at exit points can help us understand which types of exit structures are most susceptible to creating evacuation bottlenecks.

Figure 4 shows the screenshots of the pedestrian evacuation process with a 0.6 m-width exit and 500 pedestrians. For a more intuitive comparison, the depiction of crowd gathering characteristics in the exit area was standardized in all images. The crowd initially arranged itself into an arch shape at the exit. As the evacuation process advanced, the distance between the center point of the exit and the arched crowd's outermost edge gradually reduced. The red line in Figure 4a,b indicates the boundary of the fastest evacuation amongst the five scenarios at that particular moment. During the initial phase of evacuation (i.e., when the simulation time is at 100 s, as shown in Figure 1a), there is a minimal difference in the evacuation efficiency among Exits 3, 4, and 5. This efficiency is somewhat greater than what is seen at Exits 1 and 2. Nevertheless, as the evacuation progresses to its mid- and latter stages, Exits 2 and 3 demonstrate a faster evacuation rate

compared to the others. Hence, the data suggest that the evacuation strategies employed at Exits 2 and 3 are significantly more effective. One possible reason for these observed results is that, for Exit 2 and Exit 3, reaching the building' boundary signifies the end of the evacuation process. At this point, individuals are essentially no longer influenced by what is outside the boundary. In contrast, for Exits 1, 4, and 5, there are still factors like door leaves, passages, and walls outside of the building boundary that hindered the swift movement of the crowd through the building boundary.

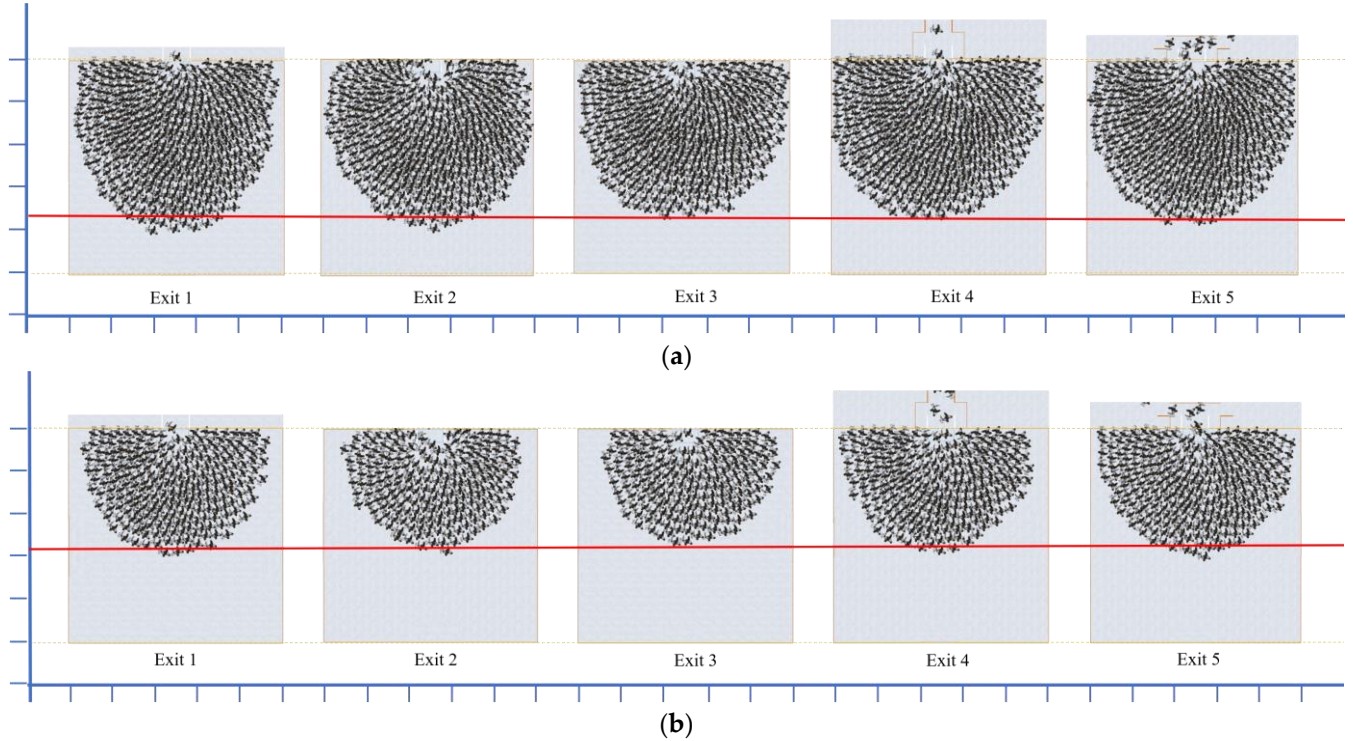

**Figure 4.** Screenshots of the pedestrian evacuation process in different exit scenarios at a certain moment. (**a**) Simulation time = 100 s; (**b**) simulation time = 300 s.

### 3.2. Evacuation Time

The evacuation time refers to the total time required for all individuals to reach a secure exterior area from inside a building under obligatory evacuation circumstances. This metric is paramount in assessing the effectiveness of evacuation plans.

Evacuation time serves as a crucial and direct indicator of evacuation efficiency. Figure 5 illustrates the variability in evacuation times across our tested scenarios, differentiated by the exit widths and crowd density levels. The curves in Figure 5 suggest that increases in the exit width result in a reduced evacuation time. Regardless of the specific crowd density and exit width, the evacuation time can vary significantly, contingent upon each individual's choice of an exit. Notably, the disparity in evacuation times among different exits appears to diminish slightly as the exit width increases. This trend of diminishing disparity in evacuation times among different exits becomes more pronounced as the population density escalates. For instance, when considering an exit width of 1.2 m and a crowd density of 1 person/m$^2$, the evacuation time spans from 99.6 s (shortest at Exit 2) to 124.8 s (longest at Exit 5), resulting in a disparity of 25.2 s. When the crowd density is amplified to 5 people/m$^2$, the evacuation time dramatically expands, ranging from 503 s (shortest at Exit 2) to 623.5 s (longest at Exit 5), establishing a more pronounced difference of 103.5 s.

Upon analyzing the time taken to reach the boundary of each evacuation area, we observed that Exit 3 generally offers the most optimal evacuation time in most circumstances, albeit with suboptimal performance in certain situations. The least efficient routes are pre-

dominantly identified at Exits 1 and 5, characterized by their prolonged evacuation times. In contrast, Exits 2 and 4 tend to yield comparable evacuation results, with closely matched evacuation times. The possible reasons for the above results are as follows. Concerning Exit 1, the presence of door leaves along the evacuation boundary may intensify competition among individuals due to their proximity to the exit boundary. Regarding Exit 5, having the exit boundary on both sides of the door may lead to collisions as people choose their exits. In the case of Exit 4, upon reaching the evacuation boundary, individuals essentially enter a relatively long corridor, promoting orderly movement among the crowd.

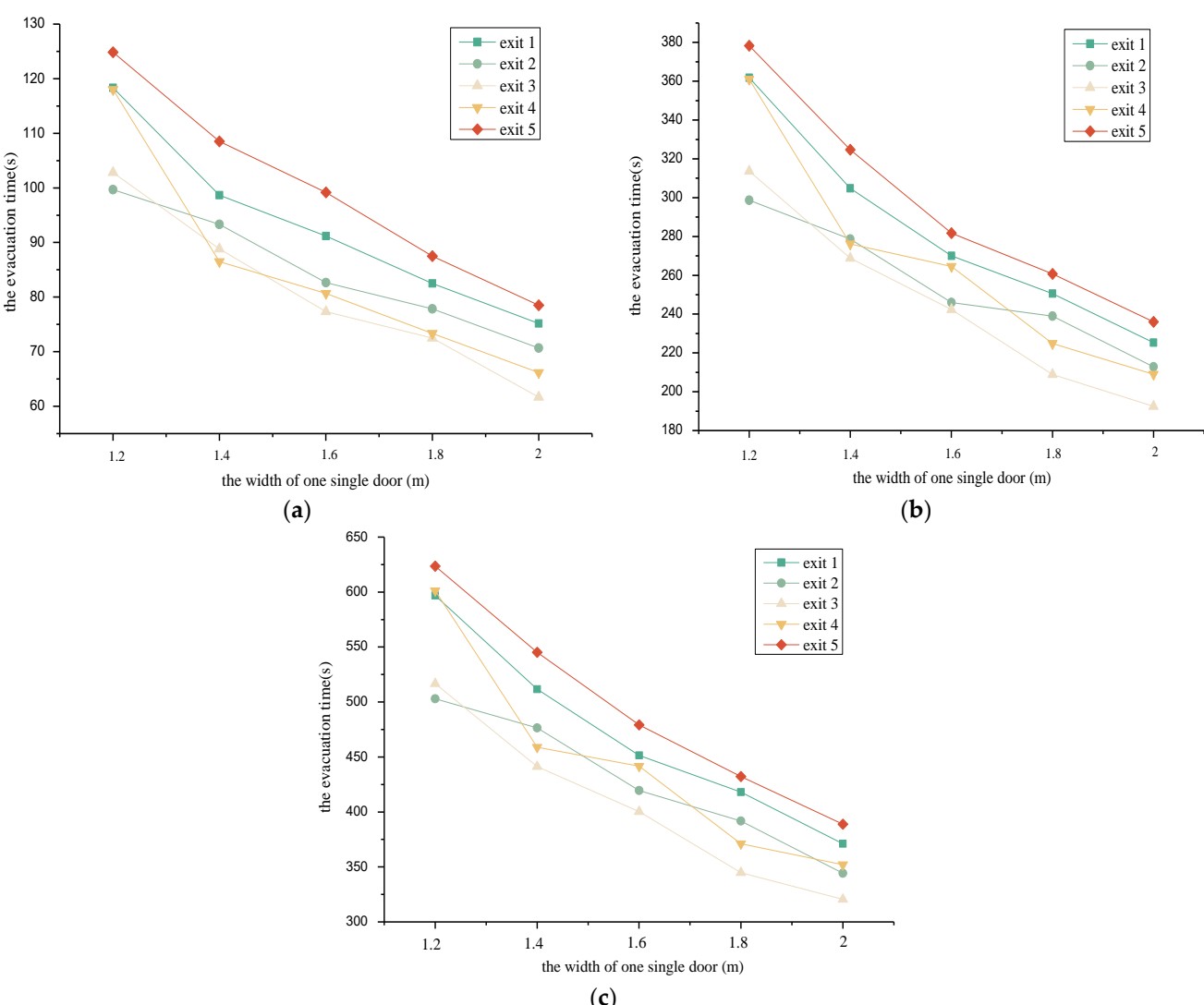

**Figure 5.** The evacuation times of the scenarios with different exits and crowd density levels. (**a**) Crowd size of 100; (**b**) crowd size of 300; (**c**) crowd size of 500.

### 3.3. Flow Rates at Different Exits

Assessing and optimizing building evacuation plans hinges on the evaluation of evacuation time and exit flow rates. Ideally, building design and evacuation planning should be oriented towards the minimization of evacuation time and the maximization of exit flow rates. The exit evacuation flow rate quantifies the number of individuals passing through a specific evacuation exit per unit time.

Figure 6 depicts the variations in the number of evacuees per unit exit width in different scenarios over time. Simulation results from various scenarios were subjected to linearly fitting, wherein the slopes of the resulting curves signify the evacuation flow rate at each exit. In Figure 6, the data reveal that the evacuation flow rates for Exits 1, 2, 3,

4, and 5 are 0.66 people/(m·s), 0.77 people/(m·s), 0.80 people/(m·s), 0.71 people/(m·s), and 0.66 people/(m·s), respectively. In alignment with these findings on evacuation time, Exit 3 exhibits the most efficient evacuation performance, closely followed by Exits 2 and 4, whereas Exits 1 and 5 manifest the lowest levels of evacuation efficiency. In our opinion, the reason for these results is related to the structural forms of the exits. Due to the presence of door leaves, different exit structures can influence the distribution of obstacles in the exit area. Additionally, various forms of exit structures may also impact the guidance of pedestrian flow in the exit area.

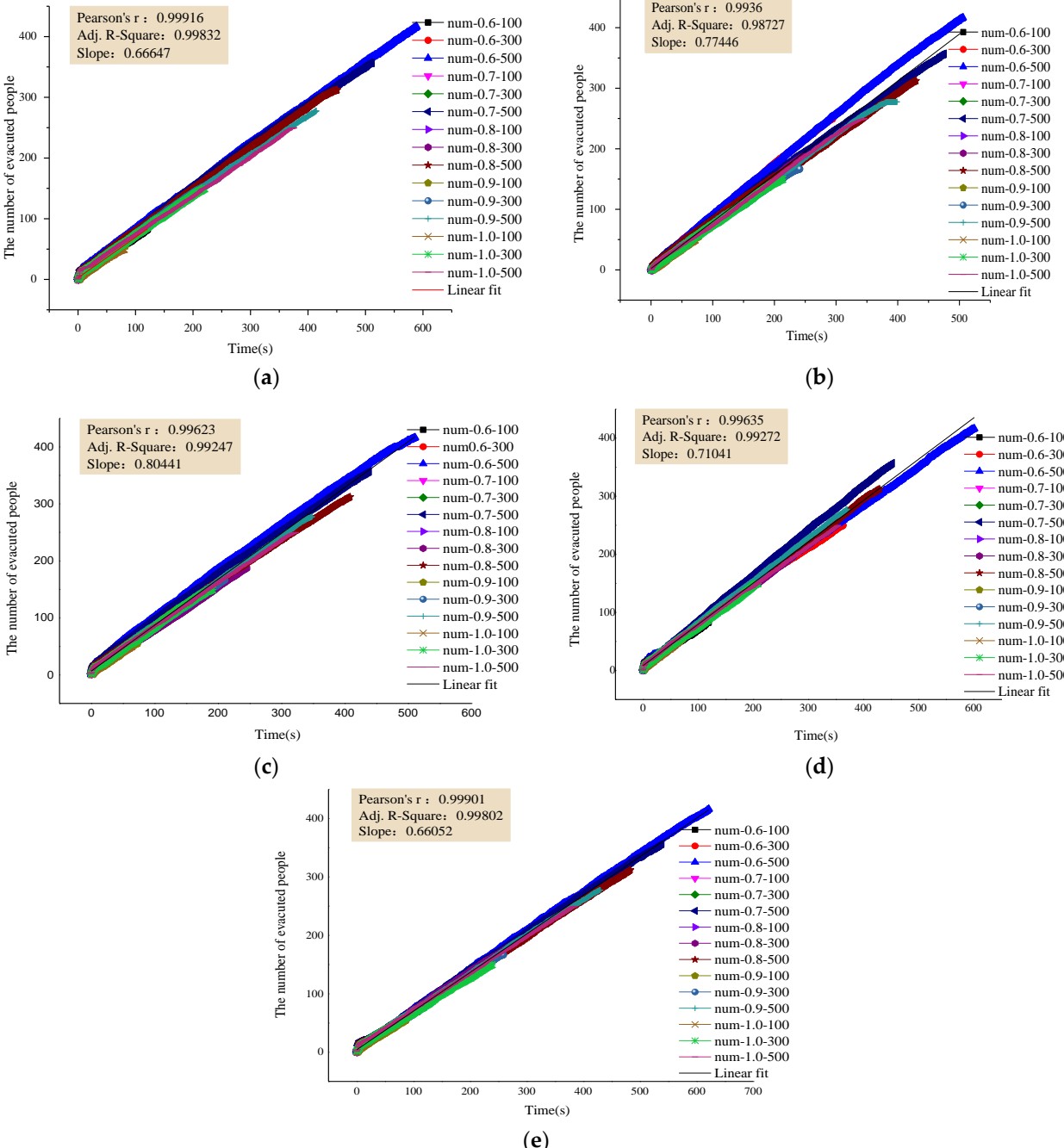

**Figure 6.** Flow rates at various exits. (**a**) Flow rate at Exit 1; (**b**) flow rate at Exit 2; (**c**) flow rate at Exit 3; (**d**) flow rate at Exit 4; (**e**) flow rate at Exit 5.

### 3.4. Crowd Density in the Observation Area

Studies on pedestrian dynamics propose that density is a principal factor influencing the efficiency of human evacuation. To ensure safety and evacuation efficiency at the exits of buildings, it is crucial to avoid excessively high crowd densities in the exit zones. The crowd density here is the number of people in the observation area divided by the area of the observation area.

As part of our analysis, we conducted an examination of variations in population density within the observation area, leading to the exit, while adjusting the exit width, as is visually represented in Figure 7. The crowd density within the observation areas for Exits 1, 3, 4, and 5 predominantly maintains a level of approximately 4 people/m$^2$. Intriguingly, these values exhibit minimal fluctuations, even with the adjustments in the exit width, indicating that exit width in isolation may not exert a significant impact on crowd density within these observation areas. In contrast, the crowd density within the observation area associated with Exit 2 consistently remains around 2 people/m$^2$. Exit 2, benefiting from this comparatively lower crowd density, facilitates a robust evacuation flow rate, rendering it more amenable to effective crowd management. A reduced crowd density contributes to smoother and more efficient evacuation procedures. This observation further reinforces the notion that strategically positioning obstacles before an exit can positively impact evacuation efficiency, as previously proposed [1].

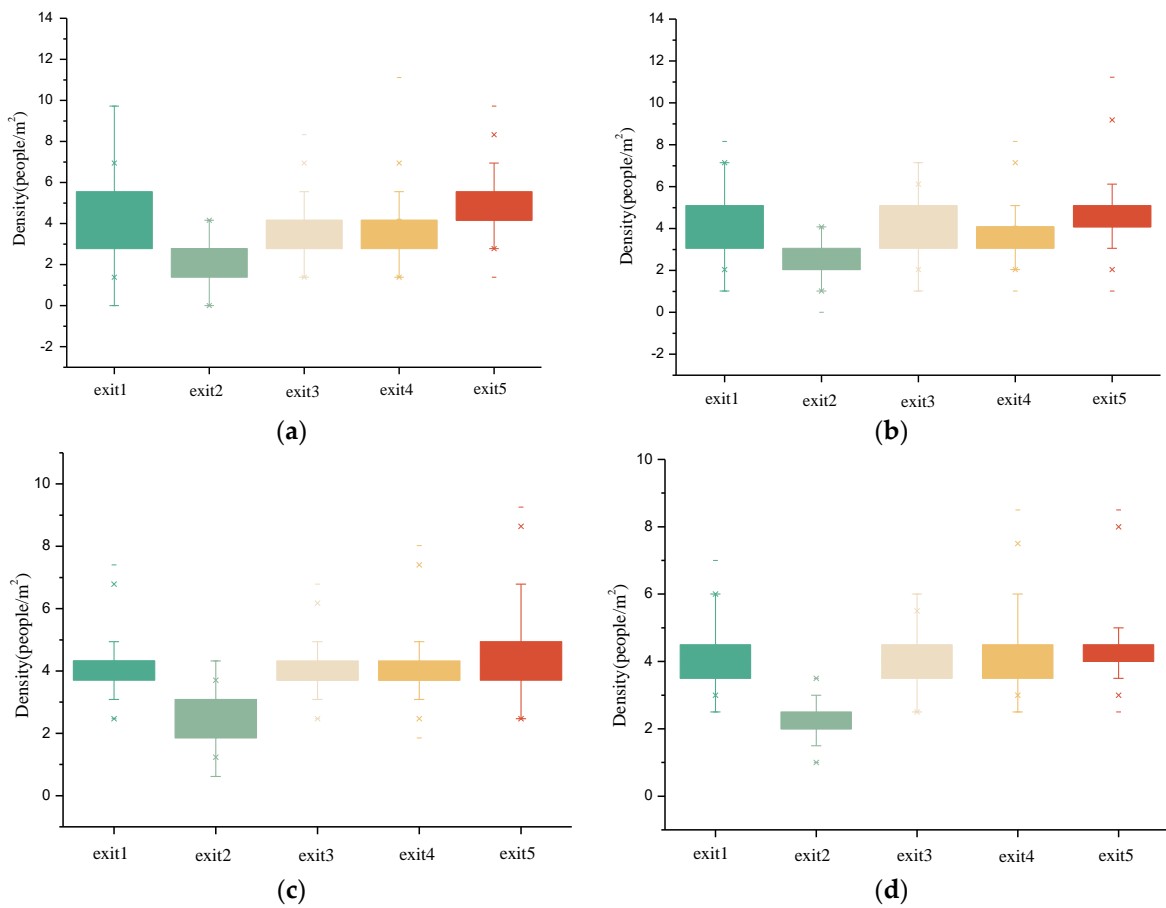

**Figure 7.** Depiction of crowd density levels within the observation area. (**a**) Exit width of 1.2 m; (**b**) exit width of 1.4 m; (**c**) exit width of 1.8 m; (**d**) exit width of 2.0 m.

Due to the inward-opening door leaves of Exit 2, when pedestrians approach the exit, the door leaves obstruct pedestrians on both sides from entering the observation area. This, to some extent, reduces the congestion of the crowd in the observation area. In our pedestrian evacuation studies, the flow of the crowd tends to increase first and

then decrease with the density of the crowd. Therefore, the case of a low density in the observation area of Exit 2, coupled with a high evacuation efficiency, is reasonable. For the same crowd density, the corresponding crowd flow is measured as a range rather than a specific value. Hence, the situation wherein the observation area density in Exit 3 is similar to that of Exits 1, 4, and 5, but with good evacuation efficiency, is also reasonable. Excessive crowd density not only affects the evacuation efficiency of exits but also increases the risk of stampede incidents. A situation with a lower crowd density and a higher crowd flow meets the safety goals we are aiming to achieve.

### 3.5. The Distribution of Time Difference

For Exits 1, 4, and 5, the arrival of the crowd at the door's perimeter effectively marks the end of the evacuation process. For Exits 1, 4, and 5, a total of 45 simulation experiments were conducted (five exit widths × three crowd densities × three repetitions). The times taken for the last pedestrian to reach the evacuation boundary (depicted by the solid green line in Figure 2) and the exit boundary (expressed by the dotted green line shown in Figure 2) were recorded for each experiment. The difference between these two times was calculated, and a frequency analysis was performed on the 45 differences. Figure 8 illustrates the time differences between the crowd's arrival at the exit boundary and the perimeter of the evacuation zone. Notably, the time differences across different exits conform closely to a Gaussian distribution. The mean and standard deviation of the time delay at Exit 1 are recorded as 2.14 s and 0.62 s, respectively. For Exit 4, the mean and standard deviation stand at 7.64 s and 2.63 s, respectively, while for Exit 5, they are 9.85 s and 2.21 s. Assessing the distribution of the evacuation time differences, it becomes evident that the arrangements at both Exit 4 and Exit 5 impede the evacuation process, thereby reducing their efficiency.

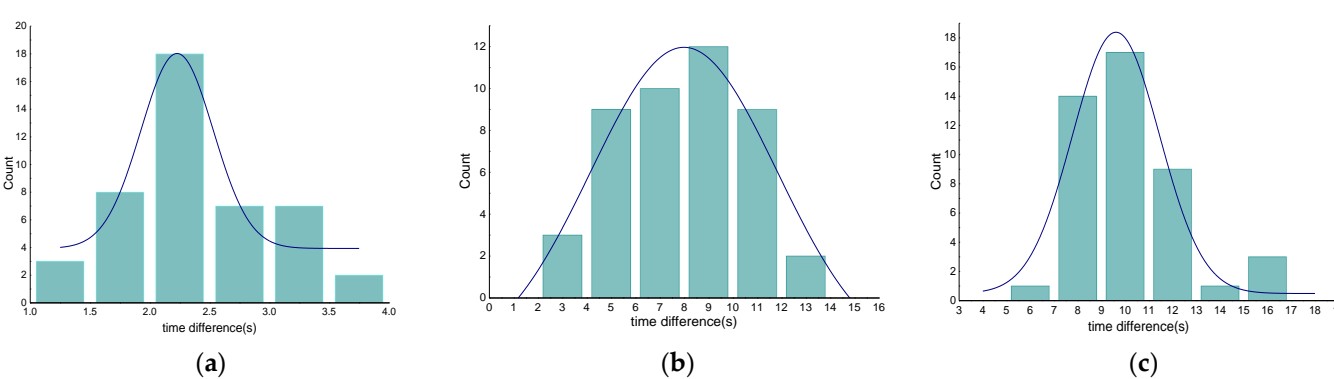

**Figure 8.** The distribution of the time differences between the crowd's arrival at both the evacuation boundary (depicted by the solid green line in Figure 3) and the exit boundary (expressed by the dotted green line in Figure 3). (**a**) Time difference at Exit 1; (**b**) time difference at Exit 4; (**c**) time difference for Exit 5.

## 4. Discussion and Conclusions

The exit structure forms a key element of any building evacuation system, significantly impacting the capacity and efficiency of crowd evacuation. The effects of structural form and the presence of attached obstacles on the evacuation efficiency of emerging commercial buildings remain unclear, posing management challenges for emergency exits. As such, this paper explored these effects by simulating crowd evacuation processes under five different exit scenarios, with Exit 3 serving as a control in this set of experiments. We analyzed key metrics, including evacuation time, flow rate, crowd density, and time differences. Our principal findings are summarized as follows:

(1)    Regarding evacuation time, Exit 3 delivered the optimal results, achieving the leading or near-top performance across various scenarios. The performances of Exits 5 and

  1 were less than ideal in comparison, while Exits 2 and 4 yielded similar evacuation outcomes.

(2) The respective flow rates for Exits 1, 2, 3, 4, and 5 were 0.66 people/(m·s), 0.77 people/(m·s), 0.80 people/(m·s), 0.71 people/(m·s), and 0.66 people/(m·s). Excluding the control experiment (Exit 3), Exit 2 proved the most efficient, with Exit 4 only slightly less so.

(3) The population density in the observation areas for Exits 1, 3, 4, and 5 mostly settled around 4 people/m$^2$, with no significant changes despite variations in the exit width. In contrast, the population density in the observation area for Exit 2 approximated 2 people/m$^2$, suggesting that the structural design of Exit 2 is preferable for controlling crowd density in the exit area.

(4) The structures of Exits 4 and 5 increased the ultimate evacuation time by 7.64 s and 9.85 s, respectively, relative to the performance of Exits 2 and 3.

  Considering all the gathered data, Exit 3 displays an outstanding evacuation efficiency, with Exit 2 also delivering notably efficient results. However, Exits 1, 4, and 5 exhibit suboptimal performance in terms of evacuation time and exit flow rates, indicating a necessity for further enhancements. It is worth emphasizing that practical building exits typically incorporate doors or gates, while Exit 3 remains confined to the realm of theoretical research and is not directly applicable to real architectural structures. In summary, Exit 2's structural design appears to offer distinct advantages in managing personnel density within the exit area, particularly when addressing concerns related to population density control.

  There are also some limitations in the current study. The micro-level effects of exit structures on crowd evacuation, such as localized crowd pressure and localized crowd flow, still need further investigation. Due to the variations in the studied exit structures, the size and scope of the exit areas also differ, making it challenging to determine a uniform study area. Building exits are crucial safety components in any structure. Their design impacts the effectiveness and speed of crowd evacuations. This study represents systematic efforts to understand and optimize evacuation processes under different building exit structures, presenting significant insights for architectural designs and building management regarding fire safety.

**Author Contributions:** X.W.: conceptualization, writing—original draft, methodology; Z.L.: conceptualization, data curation, data analysis; H.S.: writing—review and editing, funding acquisition; H.Q.: writing—review and editing; H.Y.: writing—review and editing, funding acquisition. All authors have read and agreed to the published version of the manuscript.

**Funding:** The present study was supported by the National Natural Science Foundation of China (71904006), Henan Province Key R&D Special Project (231111322200), the Science and Technology Research Plan of Henan Province (232102320043, 232102320232, 232102320046), and the Natural Science Foundation of Henan (232300420317, 232300420314).

**Institutional Review Board Statement:** This study was based on software simulations, so the ethical approval was not required.

**Data Availability Statement:** The datasets used and/or analyzed during the current study are available from the corresponding author on reasonable request.

**Acknowledgments:** The authors wish to express their appreciation to the reviewers for their valuable suggestions, which significantly enhanced the presentation of this paper. At the same time, the authors would like to extend their gratitude to Beijing Carila Tech Ltd. for providing the AnyLogic software and offering technical support. Many thanks to Shuchao Cao from the School of Automotive and Traffic Engineering, Jiangsu University for his help in building the evacuation model.

**Conflicts of Interest:** There are no conflict of interest to report regarding the present study.

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
