# Peer review of "Exploring the Impacts of Exit Structures on Evacuation Efficiency"

_fire, doi:10.3390/fire6120462_

Round 1

Reviewer 1 Report

Comments and Suggestions for Authors

An experiment was conducted to investigate five exit structures. An Anylogic software-based simulation model was used to analyze evacuation time, flow rate, crowd density, and time delay. This study offers guidance to engineering and technical professionals involved in exit design.

Although the manuscript's contents are interesting, the authors presented several numerical results without further explanation on why those happened. In other words, this manuscript presented results but does not give a discussion regarding the respective results. 

Major issues

Section 3 introduces the Anylogic software. Since this is a numerical paper, the method should be presented and discussed in this section like a reference [33]. The authors and readers can consider results more if there are numerical equations. 

Section 4.1 describes only the results without discussion. Why are Exit 2 and 3 better than the others? A difference between Exit 1 and Exit 2 is only the position of the wall, whether outside or inside. Why does this difference refine a result? What difference in the geometry makes a difference in results? 

Section 4.2 stated that the best scenario is Exit 3, followed by Exit 2 and 4 and Exit 1 and 5. Why Exits 1 and 5 was not good for evacuation? Seemingly, Exits 4 and 5 are both complex geometry, but Exit 4 is better than Exit 5. Exit 1 looks simple, but this shape is bad for evacuation. Why?

Exit 3 is simplest, and it is reasonable for fast evacuation. 

Section 4.3 quantifies the efficiency of evacuation. Depending on the scenario, the variance is large. The variance of Exits 1 and 5 is smaller than that of Exits 2, 3, and 4. Why does this happen?

Section 4.4 shows the lowest density for the Exit 2 case. Why is the density for the Exit 2 case the lowest? What geometry factor causes this difference? Moreover, Exit 3 was the best for evacuation, but its density was comparable with the other scenarios (Exits 1, 4, and 5). What is the importance of crowd density? Exit 2 has the lowest crowd density, but it does not influence evacuation time. 

Section 4.5, I could not understand how to read this graph. What is count? The caption of Figure 7 stated, 'The distribution of the time differential between the crowd's arrival at both the evacuation boundary.'

The x-axis is a time difference, and the y-axis is a count. This graph is the distribution of a count vs. a time difference. Please explain more about this graph. 

Minor issues

In the abstract, the authors summarise the research outcomes from exit 1 to exit 5. However, readers do not know their differences when they read abstracts. 

In Figure 3, respective figures need labels like (a) Exit 1 and (b) Exit 2.

Comments on the Quality of English Language

Language issues

In line 56, 'a buffer zone...'

In line 65, 'a buffer zone...'

In line 104, 'a pedestrian evacuation model in crowd evacuation scenarios.'

In line 173, '10 m by 10 m...' (insert a space between a number and unit through the manuscript)

In line 174, 'five simulation scenarios.'

Figure 2, '*' should be 'x' (a multiple mark)

In line 182, 'd x 2d' (a multiple mark. Please check through the manuscript)

In line 226, 'Figure 4...' 

In line 232, 'as the exit width increases.'

In line 246, 'evacuation plans hinges on...'

In line 271, 'in Figure 6.'

In line 363, 'Kodur, V.K.R., S. Venkatachari, an...'

Author Response

Please refer to the attachment for detailed information.

Reviewer 2 Report

Comments and Suggestions for Authors

Thank you for the opportunity to review the article, “Explore the Impact of Exit Structures on Evacuation Efficiency”. The article examines a series of evacuation simulations using the AnyLogic software to test and compare various exit structures. The article is generally well-written and contributes to the literature on evacuation efficiency.

Figure 3 would benefit from additional labeling that identifies each of the 10 exit scenarios. Are they exits 1 to 5 running left to right?

Lines 278 to 281: There is a statement about exit 2 having strategically positioned obstacles that impact evacuation efficiency. Exit 4 appears to have the same or similar barriers. The authors need to explain how the strategically positioned obstacles result in a benefit for the exit 2 scenario but not the exit 4 scenario.  

Lines 294 to 296 and 317 to 318: These two conclusions are not clear. Explain what is meant by “do not impede…efficiency”. Is there a comparison that needs to be stated. How do the longer evacuation of exits 4 and 5 show efficiency over the shorter time delay of exit 1?

Why are exits 2 and 3 not reported in Figure 7?

At the end of the conclusion, it may help to clarify the key points that support the statements about exit 2 in lines 319 to 326.   

Author Response

(The authors gave the same response as above.)

Reviewer 3 Report

Comments and Suggestions for Authors

Various designs of exit from the building are investigated to find the best one from the point of view of evacuation. This work is interesting and actual. There are some questions.

How are the results obtained related to experimental data?

In the abstract part, it is not clearly understand, what kind of exits will be considered in the paper.

The pieces of text in introduction are the same: lines 56-57 and 65-66. Moreover, the references [15] and [18] are the same.

In the section "3.1. Introduction of the Anylogic Software" it should be added references to social-force model and A*algorithm.

Line 131. The subsection name should be renamed because in this section the simulation setups are considered, not model.

Figure 2. It is not clearly understand, what "d" in figure 2(d) and figure 2(e). Are they the same or not? And, it should be added the mark where are the evacuation boundary and the exit boundary.

Lines 186-187 “To reduce the impact of randomness in the results, each scenario's evacuation simulation experiment was conducted three times.” The social-force model is a deterministic continuum model. What kind of randomness the authors reduces?

Figure 3 – is a picture with very small details. Add mark of scenarios to the figure or to the figure caption.

“4.4 Observation Area's Crowd Density” How was the density calculated? By what method?

Figure 7 requires an explanation of the experimental conditions: the width of the exit, the number of people in the evacuation area. Section “4.5 Expanded Dispersal Time Distribution” requires more explanation how these results were received.

I think, that the observation area for scenario 2 is not in the same conditions with other scenarios (1, 4, 5). Measurements are taken after the obstacle, and in other scenarios, measurements are taken before the obstacle. Perhaps this affects the result and the density in this scenario is much less than in others.

Line 363. There is a minor error in reference.

Author Response

(The authors gave the same response as above.)

Reviewer 4 Report

Comments and Suggestions for Authors

In this article, different types of exits are explored through simulation. Its effect on evacuation efficiency was studied. This work is important because it stimulates researchers to enhance the implications of relevant architectural design. However, the work is not in line with the ISI Journal, and I therefore recommend the authors to strengthen their work with further investigations as follows:

1- Although the authors stated that “the systemic research on the impact of obstacles in building exit areas on crowd evacuation regulations is still needed” the authors’ contribution did not include addressing this shortcoming. The authors recommend enhancing their work by incorporating obstacles to exits and introducing optimization solutions that improve evacuation capacity through simulation results.

2- More contributions can be combined by taking into account different situations: normal and emergency situations.

3- The authors did not clarify the configuration of the initial locations of evacuees within the evacuation area.

4- In the abstract, the authors present the following details: “the evacuation efficiency of Exit 1 and Exit 5 were relatively low. Exit 2 and Exit 4 appeared to have similar evacuation times, but Exit 2 outperformed Exit 4 in terms of evacuation flow rate and crowd density at the exit. Though Exit 3 exceled in terms of evacuation efficiency, but it is not directly applicable to real architectural structures. Therefore, Exit 2 emerged as a highly promising solution in terms of flow rate and population control in the exit area, underscoring the effectiveness and practicality of its structural design”. These details should be presented as a discussion, results, summary, or conclusion section at the end of this article. If you could rephrase these results in terms of a general description of exits characteristics and not in terms of exits 1, 2, 3...

Author Response

(The authors gave the same response as above.)

Round 2

Reviewer 1 Report

Comments and Suggestions for Authors

Although the authors addressed my concerns, most of those only appear in the cover letter. As I mentioned, the manuscript presented results, but discussions were few. I recommended the authors add discussions based on the cover letter in different paragraphs. Answers 3-5 can be discussion paragraphs; the authors can put those in respective sections. 

Answer 4: 

Some lines are slightly away from the fitted lines, meaning that 'variance' is 'Adj—r-Square' of Exits. For example, the R-square value of Exit 2 is relatively lower than in that of the other cases. Is there a physical meaning?

Please check whether 'micro-level' is the correct word or not. Readers may think that micro-level means 10^-6 m order. 

Comments on the Quality of English Language

Language issues

On p. 10 in lines 305-306, Figure 2 (remove dot) 

Please insert a space between a number and a unit, for example, 99.6 s and 2 people/m2.

There are many * instead of a multiple mark. Please replace those.

Besides, * is used instead of a dot, for example, lines 19-20 in the abstract. 

Reviewer 3 Report

Comments and Suggestions for Authors

1. Check the reference to Figure 2 in the caption to Figure 8:  "Figure 8. The distribution of the time differential between the crowd's arrival at both the evacuation boundary (depicted by the solid green line in Figure 2) and the exit boundary (expressed by the dotted green line shown in Figure 2). (a) Time differential at Exit 1; (b) Time differential at Exit 4; (c) Time differential for Exit 5."

2. The article should have a section with a conclusion or discussion and  conclusion, not just discussion.
